# Comprehensive transcriptomic analysis of prostate cancer lung metastases

**Alireza Saraji**[1]©, **Katharina Wulf**[1]©, **Janine Stegmann-Frehse**[1], **Duan Kang**[2], **Anne Offermann**[3], **Danny Jonigk**[4], **Mark Philipp Kuehnel**[5,6], **Jutta Kirfel**[1], **Sven Perner**[7], **Verena Wilbeth Sailer**[1]*

**1** Pathology of the University Hospital Schleswig-Holstein, Campus Luebeck, Luebeck, Germany,
**2** Guangdong Second Provincial General Hospital, Guangzhou, P. R. China, **3** Institute of Pathology, University of Muenster, Muenster, Germany, **4** Institute of Pathology, RWTH Aachen, Aachen, Germany, **5** Institute of Pathology, Hannover Medical School, Hannover, Germany, **6** Biomedical Research in Endstage and Obstructive Lung Disease (BREATH), German Center for Lung Research, Hannover, Germany, **7** Institute for Hematopathology, Hamburg, Germany

© These authors contributed equally to this work.
* Verena-Wilbeth.Sailer@uksh.de

**Data Availability Statement:** All relevant data are within the manuscript and its Supporting information files.

## Abstract

Metastatic prostate cancer (mPCa) is a widespread disease with high mortality. Unraveling molecular mechanisms of disease progression is of utmost importance. The microenvironment in visceral organs and the skeletal system is of particular interest as a harbinger of metastatic spread. Therefore, we performed a comprehensive transcriptomic analysis of prostate cancer lung metastases with a special focus on differentially expressed genes attributable to the microenvironment. Digital gene expression analysis using the NanoString nCounter analysis system was performed on formalin-fixed, paraffin-embedded (FFPE) tissue from prostate cancer (PCa) lung metastases (n = 24). Data were compared to gene expression data from primary PCa and PCa bone metastases. Bioinformatic analysis was performed using several publicly available tools. In comparison to prostate cancer bone metastases, 209 genes were significantly upregulated, and 100 genes were significantly downregulated in prostate cancer lung metastases. Among the up-regulated genes, the top 10 genes with the most significant P-value were HLA-DPB1, PTPRC, ITGB7, C3, CCL21, CCL5, ITGAM, SERPINA1, MFAP4, ARAP2 and among the down-regulated genes, the top 10 genes with the most significant P-value were FOXC2, TWIST1, CDK14, CHAD, IBSP, EPN3, VIT, HAPLN1, SLC44A4, TBX1. In PCa lung metastases genes associated with immunogenic responses were upregulated while genes associated with epithelial-mesenchymal transition were down-regulated. We also showed that CXCR3/CXCL10 axis plays a significant role in prostate cancer lung metastases in comparison to bone metastases. In this study, we comprehensively explored transcriptomic alterations in PCa lung metastases in comparison to primary PCa and PCa bone metastases. In PCa lung metastases genes associated with immunogenic responses are upregulated while genes associated with epithelial-mesenchymal transition are down-regulated. This points to a more immunogenic phenotype of PCa lung metastases thus potentially making patients more susceptible to immunotherapeutic approaches.

**Funding:** This work was funded by the German Research Foundation (DFG) with the priority program µBone number SPP2084 (to VS and SP). The funders had no role in study design, data collection and analysis, decision to publish, or preparation of the manuscript.

**Competing interests:** The authors have declared that no competing interests exist.

**Abbreviations:** DGE, digital gene expression; EMT, epithelial to mesenchymal transition; FFPE, formalin-fixed paraffin-embedded; mPCa, metastatic prostate cancer; TAM, tumour-associated macrophages.

## Introduction

Metastatic prostate cancer places a heavy burden on the individual patient, his next of kin, and the health care system. An estimated 34.310 men had been predicted to die of prostate cancer in 2021 [1]. The most common metastatic sites are bone, lung, and liver [2]. Metastatic spread is a complex multistep process for which the seed and soil concept introduced by Steven Paget in 1889 still holds. Since Dr. Paget's time, the concept has undergone significant expansion [3, 4]. Numerous contributing factors both in the seed, i.e. the tumour cell as well as in the soil, the organ in which metastatic disease occurs have since been identified [4]. In a large study that employed data from the Surveillance, Epidemiology and End Results (SEER) database, the incidence of lung metastasis (LM) in all prostate cancer patients was low (0.43%). However, the presence of LM significantly increased the risk of cancer-related death with a hazard ratio of 16 [5]. Lung metastases not only shorten a patient's lifespan significantly, but they also result in severe morbidity and impaired quality of life due to cough, shortness of breath, and pleural effusion [6]. There is thus an urgent clinical need to prevent metastatic disease and to formulate potent anti-tumour strategies in patients with prostate cancer and lung metastases. Understanding the fertile soil might contribute to preventing tumour cells from forming overt and clinically relevant metastatic lesions. To add to the growing body of knowledge regarding the microenvironment we performed a comprehensive gene expression analysis of prostate cancer lung metastases.

## Materials and methods

All methods in this study were performed in accordance with the relevant guidelines and regulations approved by University of Luebeck and University Hospital Schleswig-Holstein (UKSH) Luebeck.

### Ethics

This study was approved by the Ethics Committee of the University of Luebeck (project code 18–053, date of approval: March 2nd, 2018, date of amendment: June 17th, 2020).

### Cohort

Formalin-fixed and paraffin-embedded (FFPE) archival material from prostate cancer bone metastases (30 cases) and prostate cancer lung metastases (29 cases) along with primary prostate cancer (28 cases) as reference were selected from the archives of the Institute of Pathology, University Hospital Schleswig-Holstein (UKSH) Luebeck and the Research Center Borstel, Germany (Table 1). According to our ethics vote the FFPE samples were accessed for research purposes from March 2nd, 2018. Diagnosis of prostate cancer lung metastases was confirmed by a board-certified pathologist. Both primary and metastatic tumour samples were available for two patients. This study was approved by the Ethics Committee of the University of Luebeck (project code 18–053, date of approval: March 2nd, 2018, date of amendment: June 17th, 2020).

### mRNA extraction

Hematoxylin and eosin-stained slides were evaluated by a pathologist (VS, AO) for tumour cell content and subsequently annotated for macrodissection. mRNA extraction was performed according to standard protocols and as described before [7, 8].

Briefly, the Paraffin blocks were sectioned into 8 µm cuts and each slide was compared with the annotated HE slide. Marked cancer tissue was scraped off with a scalpel and transferred

**Table 1. Patient's (FFPE) demographic table.**

| Material | Number of patients (total FFPE) | Age at Biopsy (in years / %) | | Year of Biopsy (in years / %) | | Gleason Score | |
|---|---|---|---|---|---|---|---|
| **Primary Prostate Cancer** | 28 | <60 | 4 (14%) | 2000–2005 | 27 (96%) | <7 | 8 (29%) |
| | | 60–65 | 7 (25%) | | | | |
| | | 66–70 | 10 (36%) | | | >7 | 19 (68%) |
| | | 71–75 | 6 (21%) | 2006–2010 | 1 (4%) | | |
| | | >75 | 1 (4%) | | | unknown | 1 (4%) |
| **Lung Metastasis** | 29 | <66 | 0 | 2000–2005 | 3 (10%) | Not available | |
| | | 66–70 | 7 (24%) | 2006–2010 | 2 (7%) | | |
| | | 71–75 | 3 (10%) | 2011–2015 | 4 (14%) | | |
| | | >75 | 13 (45%) | 2016–2020 | 12 (41%) | | |
| | | unknown | 6 (21%) | unknown | 8 (28%) | | |

directly to the lysis buffer into an RNAase-free tube. RNA was isolated using the automatic bead-based Maxwell RSC RNA FFPE Kit (Cat. No.: AS1440, Promega) according to the manufacturer's instructions. The RNA was eluted in water and then measured with Qubit. The RNA samples were divided into 7 μl aliquots and stored at −80˚C. Extracts with RNA concentrations of at least 10 ng/μl and sufficient RNA integrity with at least 90% of the fragments longer than 100 nucleotides were considered as suitable for gene expression analysis.

## Digital gene expression analysis (DGE)

DGE using the Nanostring platform was performed as described before [7, 8]. A commercially available gene panel (nCounter PanCancer Progression Panel) consisting of 770 genes implicated in cancer progression and metastasis was employed. This gene panel consists of 277 angiogenesis-, 254 extracellular matrix-, 269 epithelial-mesenchymal transition, and 173 metastasis-related genes [9]. Data were analysed using the nSolver Advanced Analysis software provided by Nanostring.

## Bioinformatical analysis and data platform creation (MACNP-L)

To assess the biological function of all differentially expressed genes a data platform (MACNP-L, "Manually Annotated and Curated NanoString-data Platform for Lung metastases") was created. 312 differentially expressed genes were manually annotated. MACNP-L contains information about the biological process, molecular function, protein class, pathways, synonyms, and available publications for all of the differentially expressed genes. Several advanced bioinformatic tools were manually applied for data enrichment and annotation. In particular, PANTHER (pantherdb.org) for information about the biological process, KEGG PATHWAY database (genome.jp/keg) for information about the pathways, and Metascape (metascape.org) for information about the molecular function were consulted for this purpose. The DEGs protein-protein network (PPI) was constructed from STRING (https://string-db.org) following with MCODE plugin Cytoscape (https://cytoscape.org) to calculate the key sub-clusters in the entire PPI network and the seed genes in each sub-cluster. PubMed (pubmed.ncbi.nlm.nih.gov) was searched for existing publications regarding genes in context with prostate cancer and metastases [10–17].

## Statistical analysis

An independent sample t-test was used to compare differences in mRNA counts in prostate cancer lung metastases and prostate cancer bone metastases. DEGs with Log2 fold change > 0

are considered meaningful difference results. Default parameters were set for calculation when STRING and MCODE were used. P value <0.05 was considered statistically significant. IBM SPSS Statistics version 2.0 was used for statistical validation and R language (version 3.6.3) was used for graphing and data presentation. All the sample sizes are mentioned in each figure.

## Results

### Gene expression analysis results

24 of 29 samples provided mRNA sufficient in quantity and quality to successfully perform digital gene expression analysis (DGE). Setting prostate cancer bone metastases as a reference we found that 209 genes were significantly upregulated, and 100 genes were significantly downregulated in prostate cancer lung metastases (S1 & S2 Tables). Among the up-regulated genes, the top 10 genes with the most significant P-value are HLA-DPB1, PTPRC, ITGB7, C3, CCL21, CCL5, ITGAM, SERPINA1, MFAP4, ARAP2 and among the down-regulated genes, the top 10 genes with the most significant P-value are FOXC2, TWIST1, CDK14, CHAD, IBSP, EPN3, VIT, HAPLN1, SLC44A4, TBX1 (Fig 1).

### NCAM1 pitfall

NCAM1 (CD56) was significantly (p-value < 0.05) downregulated in prostate cancer lung metastases. This finding was not unexpected since CD56 is physiologically expressed on osteoblasts, which do not occur in lung tissue [18]. This illustrates that in interpreting "omics'" data from different anatomical locations obtained by macrodissection care must be taken to consider the normal cellular compartment as well so as not to be led to false conclusions. The same holds true for genes associated with ossification identified by functional enrichment analysis, that were also downregulated in lung cancer metastases.

### Altered pathways and biological processes in the metastatic microenvironment

Bioinformatics analysis using Metascape for the key sub-cluster functional enrichment performed and represents chemotactic genes as the most prominent biological functions of lung metastasis (Fig 2a). Our further analysis using STRING shows CXCL13 in the protein-protein interactome network of DEGs as the main seed of the network with highest MCODE score (Fig 2b).

**Regulation of chemotaxis.** CD36 was highly and significantly (p-value < 0.05) upregulated in prostate cancer lung metastases in comparison to prostate cancer bone metastases. CD36 is expressed on macrophages and alveolar macrophages are part of normal lung tissue. It can also be found on endothelial cells [19]. Tumour-associated macrophages (TAM) are well known in cancer and can promote invasiveness and epithelial to mesenchymal transition (EMT) [4]. Given the more than 2fold increase of CD36 expression in prostate cancer lung metastases it can be hypothesized that CD36-expressing macrophages play a genuine role in the metastatic niche of the lung. In addition, we found that C-C chemokines were also highly and significantly (p-value < 0.05) upregulated in prostate cancer lung metastases in comparison to prostate cancer bone metastases. These molecules play a role in recruiting macrophages and in TAM. CCL5 is a pro-inflammatory chemokine that is highly expressed on resting TAM and influences major intracellular signaling cascades [20, 21]. It is expressed by T-lymphocytes and macrophages but also secreted by tumour cells [22]. CCL5 belongs to the family of C-C chemokines as do CCL7, CCL8, and CCL21, which were also upregulated in prostate cancer lung metastases. They can exert both pro- and anti-tumour properties via their receptors

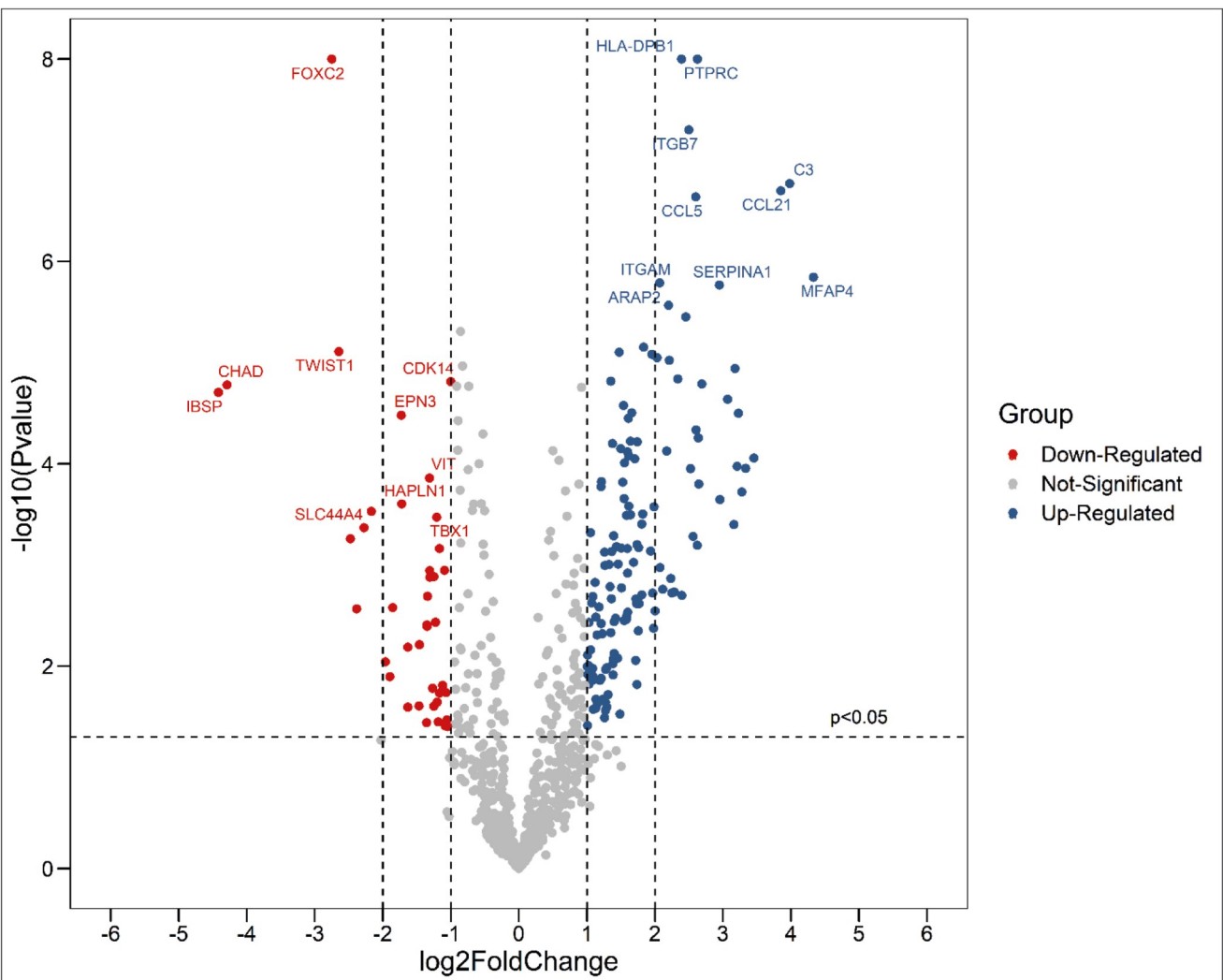

**Fig 1. Volcano plot of DEGs between lung and bone metastases.** Red dots represent down-regulated genes in lung metastases, blue dots represent up-regulated genes in lung metastases and grey dots represent genes with insignificant P-values.

CCR1 and CCR2 and prepare the ground for the inflammatory component in the tumour microenvironment [23]. TAM recruitment is particularly important and CCL7 and CCL8 can also attract tumour-infiltrating lymphocytes [23–25]. So far it has not been known that these C-C chemokines are implicated in facilitating prostate cancer lung metastases. CCL21 has been shown to promote prostate cancer cell migration [26]. CCL21 seems to be implicated in the microenvironment of primary prostate cancer as well [27]. In our dataset it was 14,42 fold upregulated in lung metastases in comparison with bone metastases. In addition, CCR2 was highly upregulated in lung metastases. CCR2 expression has been shown to correlate with prostate cancer progression [28, 29]. CCL2, one of the ligands of CCR2 contributes to chemoresistance against taxanes in a cabazitaxel-resistant cell line [30]. A CCR2 antagonist in combination with cabazitaxel increases apoptosis in the aforementioned cell line (Fig 3).

**Chemokine activated pathways.** Several C-X-C motif (CXC) chemokines were upregulated in prostate cancer lung metastases, notably CXCL10, CXCL11, and CXCL13 (Figs 2b and 3). These chemokines are thought to have angiostatic properties [31]. They also recruit tumour-

(a)

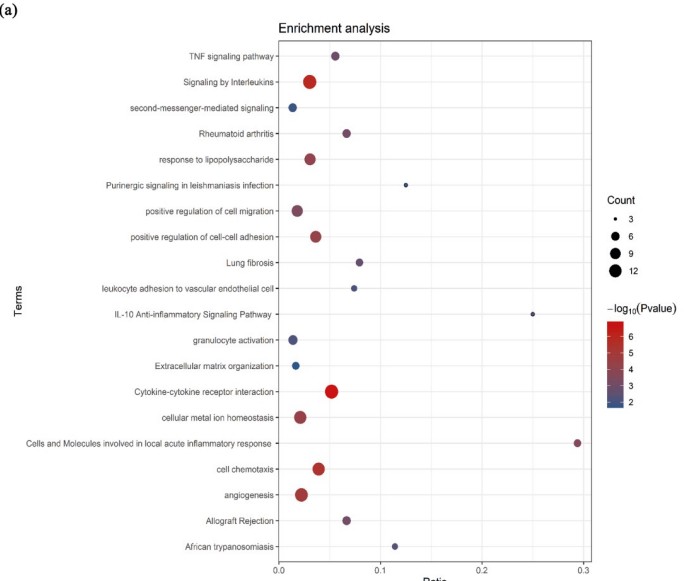

(b)

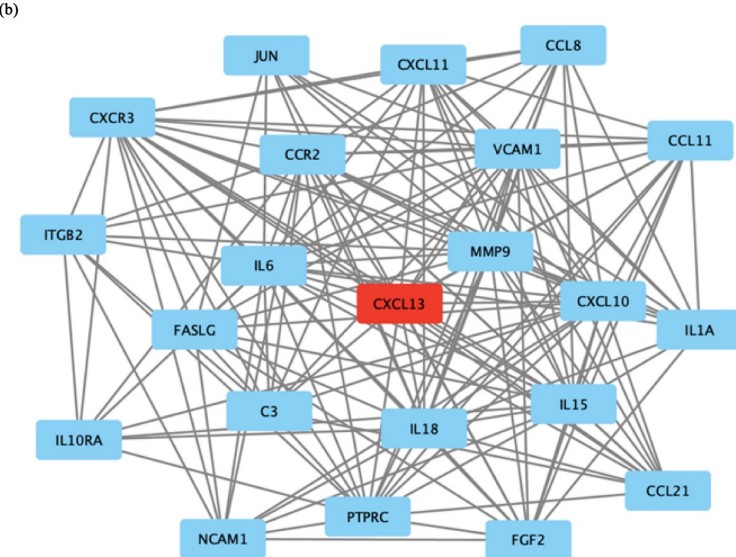

**Fig 2. Altered pathways and biological processes in the metastatic microenvironment. (a)** Bubble plot of the key sub-cluster functional enrichment created by Metascape represents chemotactic genes as the most prominent biological functions of lung metastasis **(b)**. Protein-protein interactome network of DEGs representing sub-clusters with the highest MCODE scores. CXCL13 is the seed gene in this sub-cluster (Log2FC = 1.29). Created with STRING (https://string-db.org) and Cytoscape (https://cytoscape.org).

infiltrating lymphocytes, regulatory T-cells, and myeloid-derived suppressor cells into the meta-static niche [31]. CXCL10 and CXCL11 contribute their angiostatic effects via the receptor CXCR3, which was also highly upregulated in lung metastases [31, 32]. CXCR3 has been shown to be downregulated in prostate cancer bone metastases in comparison with lymph node metastases. CXCL10 and CXCL11 as well as CXCR3 are upregulated in lymph node-positive primary prostate cancer in comparison to lymph node-negative prostate cancer [33]. The CXCR3/CXCL10 axis seems to be involved in metastatic disease in several tumour types [34].

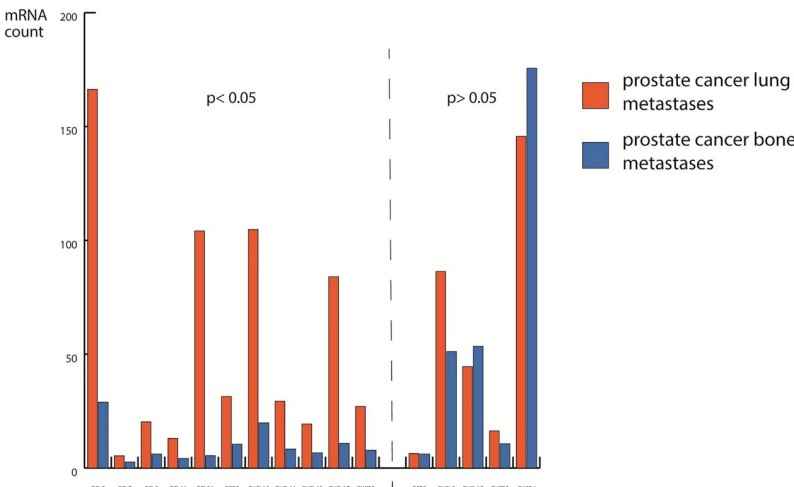

**Fig 3. Differentially regulated C-C motif in the metastatic microenvironment.** Chemokine ligands (CCL), their receptors (CCR), CXC motif chemokine ligands (CXCL), and their receptors (CXCR) in prostate cancer lung (orange) and bone (blue) metastases.

**EMT pathways are downregulated in PCa lung metastases.** Genes associated with EMT were found to be downregulated in PCa lung metastases in comparison to PCa bone metastases. E.g., the expression of Forkhead Box Protein C2 (FOXC2) was significantly lower in PCa lung metastases. FOXC2 represses E-cadherin transcription thus contributing to EMT [35]. This results in a more aggressive phenotype. High expression of FOXC2 is associated with a shorter time to biochemical recurrence after radical prostatectomy for Gleason-Score 7 PCa [36]. Interestingly and congruent with our data, the same authors found a lower expression of FOXC2 protein in non-skeletal metastases in comparison to skeletal metastases [36]. TWIST1, another well-known transcription factor implicated in EMT was also downregulated in our cohort [37].

## Discussion

We have performed comparative gene expression analysis and integrative data analysis of prostate cancer bone and lung metastases to unravel the unique transcriptome of the pulmonary metastatic niche. To the best of our knowledge, this is the first comparative analysis of its kind in PCa lung metastases. We carefully interpreted our data in the context of each anatomical location to allow for specific gene expression patterns that are normal in lung and bone, e.g., higher expression levels of genes associated with ossification in bone samples. Several differentially modified pathways are highlighted in this manuscript. In general, genes associated with immune responses are upregulated in PCa lung metastases while genes associated with EMT are downregulated. If hematogenous spread to the lungs is simply a stochastic effect tumour cells would not need a more pronounced mesenchymal phenotype other than the capacity to invade blood vessels. It is more likely though that the distinct properties by which aggressive PCa cells are preparing the pulmonary pre-metastatic niche have yet not been unraveled. In addition, our study looked at already established metastases. PCa cells in the lung might profit from a more immunogenic phenotype for successful and sustained growth. We were able to show that TAM and the chemokines that attract them play an important role in PCa lung metastases. This might open new opportunities both for diagnostic purposes as

well as for therapy. Circulating chemokines can be measured in the serum and might indicate, that lung tissue is been prepared as a seeding ground for prostate cancer cells [38]. Interestingly, *CDK12* mutant prostate cancers with loss of *CDK12* represent a distinct class of prostate cancer. They are among other features characterized by an increase of CCL21 in the microenvironment and *CDK12* mutant tumours might benefit from immunotherapy [39]. Therefore CCL21 might serve as a surrogate parameter to identify patients who are suitable for immunotherapy. Given its high increase in prostate cancer lung metastases, it might also be an attractive therapeutic target. However, since CCL21 and its receptor CCR7 show both anti- and protumour effects, it is still unclear how to best exploit the upregulation of CCL21 therapeutically [40]. The difference in the immune microenvironment between prostate lung and bone metastases is intriguing, suggesting an even greater role in preparing the pulmonary niche for metastatic disease. Therefore, these patients might profit from immune-targeting therapies.

The CXCR3/CXCL10 axis has been known for its pro-metastatic potential and is highly expressed in several tumour types. Here we show that this chemokine axis plays a significant role in prostate cancer lung metastases in comparison to bone metastases. The underlying biological reason for this is unclear and the data regarding pro- or antitumour effects are conflicting [41]. Wightman et. al have pointed out that we must look beyond canonical effects of CXCR3/CXCL10 such as the attraction of immune cells and angiogenesis, especially when dealing with autocrine stimulation. Further research is needed to unravel the interplay between the CXCR3/CXCL10 axis, the metastatic niche as well as the tumour cell. However, CXCR3 and its ligands might become attractive therapeutic targets. Inhibiting CXCR3 in preclinical models resulted in inhibition of lung metastases in several tumour types (breast, colon, osteosarcoma) [42, 43].

There are some limitations to our study. Prostate cancer metastases are rarely resected and therefore the number of cases in this study is limited. We also have very limited clinical data, especially regarding anti-androgen treatment and castration status. Since this study focuses on the organ-specific microenvironment, the lack of clinical data is of lesser importance. However, since treatment itself changes the molecular landscape of metastatic disease it stands to reason that the microenvironment also undergoes evolution. This is quite obvious in the case of neuroendocrine prostate cancer which has a predilection for forming bulky disease and causing visceral metastases. A future study with a higher number of cases and incorporating extensive clinical data might address the pending question of treatment-related alterations in the metastatic niche.

## Conclusion

In this study, we comprehensively explored transcriptomic alterations in PCa lung metastases in comparison to primary PCa and PCa bone metastases. In PCa lung metastases genes associated with immunogenic responses are upregulated while genes associated with epithelial-mesenchymal transition are down-regulated. This points to a more immunogenic phenotype of PCa lung metastases thus potentially making patients more susceptible to immunotherapeutic approaches.

## Supporting information

**S1 Table. Comprehensive excel file represented the complete list of all up and down regulated genes with detailed annotation for each gene.**
(PDF)

**S2 Table. Excel file represented the complete list of all up and down regulated genes calculated with FDR or adjusted p-Value.**
(PDF)

**S1 File.**
(PDF)

**S2 File.**
(PDF)

## Acknowledgments

We thank Eva Dreyer for her technical assistance.

## Author Contributions

**Conceptualization:** Anne Offermann, Jutta Kirfel, Sven Perner, Verena Wilbeth Sailer.

**Data curation:** Alireza Saraji, Katharina Wulf, Duan Kang, Anne Offermann, Verena Wilbeth Sailer.

**Formal analysis:** Alireza Saraji, Katharina Wulf, Janine Stegmann-Frehse, Duan Kang, Danny Jonigk, Mark Philipp Kuehnel, Verena Wilbeth Sailer.

**Funding acquisition:** Jutta Kirfel, Sven Perner, Verena Wilbeth Sailer.

**Investigation:** Alireza Saraji, Anne Offermann, Jutta Kirfel, Sven Perner, Verena Wilbeth Sailer.

**Methodology:** Alireza Saraji, Katharina Wulf, Janine Stegmann-Frehse, Duan Kang.

**Project administration:** Janine Stegmann-Frehse, Jutta Kirfel, Sven Perner, Verena Wilbeth Sailer.

**Resources:** Anne Offermann, Danny Jonigk, Jutta Kirfel, Sven Perner, Verena Wilbeth Sailer.

**Software:** Alireza Saraji, Katharina Wulf, Duan Kang, Danny Jonigk, Mark Philipp Kuehnel, Verena Wilbeth Sailer.

**Supervision:** Verena Wilbeth Sailer.

**Validation:** Alireza Saraji, Katharina Wulf, Janine Stegmann-Frehse, Duan Kang.

**Visualization:** Alireza Saraji, Katharina Wulf, Janine Stegmann-Frehse, Duan Kang.

**Writing – original draft:** Alireza Saraji, Katharina Wulf, Verena Wilbeth Sailer.

**Writing – review & editing:** Alireza Saraji, Katharina Wulf, Anne Offermann, Danny Jonigk, Mark Philipp Kuehnel, Verena Wilbeth Sailer.

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
