## [Decision Letter · Decision Letter 0]

6 Mar 2024

PONE-D-23-41725Comprehensive transcriptomic analysis of prostate cancer lung metastasesPLOS ONE

Dear Dr. Saraji,

Thank you for submitting your manuscript to PLOS ONE. After careful consideration, we feel that it has merit but does not fully meet PLOS ONE’s publication criteria as it currently stands. Therefore, we invite you to submit a revised version of the manuscript that addresses the points raised during the review process.

The study on prostate metastases gene expression profiling using FFPE samples is critiqued for not incorporating control specimens, a step crucial for identifying uniquely expressed genes between bone and lung metastases. Adjusted p-values are recommended for precise identification of differentially expressed genes, minimizing false positives. Additionally, standardizing mRNA quantification using RPKM/FPKM is suggested to ensure data reliability. The method for assessing RNA integrity, based on fragment length, is deemed inadequate; calculating the RNA Integrity Number (RIN) is proposed for more accurate quality assessment. The article's low-resolution figures also require improvement for better interpretation. Further suggestions include adding a patient demography table for contextual clarity, providing explicit criteria for mRNA regulation cut-offs, validating top dysregulated genes through secondary techniques like qRT-PCR, and offering a clearer rationale for selecting chemotactic genes in the analysis of altered pathways. These recommendations aim to bolster the study's methodological rigor, clarity, and the comprehensiveness of its findings on the gene expression profiles of prostate metastases.

We look forward to receiving your revised manuscript.

Kind regards,

Sumit Kumar Hira, Ph.D.

Academic Editor

PLOS ONE

Journal Requirements:

2. Thank you for stating the following financial disclosure: "This work was funded by Deutsche Forschungsgemeinschaft (DFG) Schwerpunktprogramm µBone SPP2084 (to VS and SP). Duan Kang was supported by China Scholarship Council at the University of Luebeck (No. 202008440263)."

Reviewers' comments:

Reviewer's Responses to Questions

**Comments to the Author**

1. Is the manuscript technically sound, and do the data support the conclusions?

Reviewer #1: Partly

Reviewer #2: Yes

2. Has the statistical analysis been performed appropriately and rigorously? 

Reviewer #1: No

Reviewer #2: Yes

3. Have the authors made all data underlying the findings in their manuscript fully available?

Reviewer #1: Yes

Reviewer #2: No

4. Is the manuscript presented in an intelligible fashion and written in standard English?

Reviewer #1: Yes

Reviewer #2: Yes

5. Review Comments to the Author

Reviewer #1: The study by Saraji et al. demonstrated the comprehensive transcriptomic analysis of prostate cancer lung metastases. The authors should modify the manuscript by considering following points.

1.The authors have used FFPE samples of prostate metastases samples and compared transcriptome profile of bone metases and lung metases samples.But authors have not used any control/adjacent control specimens to derive the specific differentially expressed genes between the subsets. Justify.

2.Regarding the statistical analysis of gene expression data, adjusted p value should be calculated to identify the differentially expressed genes in each category of samples.

3. mRNA count should be expressed as RPKM/FPKM which is the standard protocol for RNA sequencing data.

4.Here, RNA integrity with at least 90% of the fragments longer than 100 nucleotides were considered as suitable for gene expression analysis. Instead of the parameter, RIN should be calculated for the samples which is the standard protocol for gene expression analysis.

5.The resolution of the figures are very poor. That should be modified.

Reviewer #2: In this manuscript, the authors have comprehensively explored transcriptomic alterations in PCa lung metastases in comparison to primary PCa and PCa bone metastases. They have reported that upregulation of immunogenic response genes and downregulation of epithelial-mesenchymal transition genes were associated with PCa lung metastases, thereby, suggesting that lung metastases from PCa have a higher immunogenic character, which may increase patients' susceptibility to immunotherapeutic treatments.

The structuring of the manuscript is well-conceived, easy to follow, and the literature survey in the related field is comprehensive. However, the reviewer found some ways to do the manuscript better for the publication purpose.

The comments from the reviewer are listed below:

Comment 1:

The article should include a patient demography table.

Comment 2:

It is not clear from the manuscript which log2 Fold cut-off value has been considered for determining mRNA up- and down-regulation. Additionally, the author should include a table or a supplementary Excel file listing all upregulated and downregulated genes.

Comment 3:

The dysregulated candidate gene expression (Top 10 up and down) should have been validated by a secondary technique (say qRT-PCR) in the same sample-set.

Comment 4:

The authors can be clearer in explaining the choice of chemotactic genes in the result section under the subheading "Altered pathways and biological processes in the metastatic microenvironment," since Figure 2a shows additional altered pathways and biological processes in the metastatic microenvironment.

6. PLOS authors have the option to publish the peer review history of their article (what does this mean?). If published, this will include your full peer review and any attached files.

Reviewer #1: No

Reviewer #2: No

---

## [Author Response · Author response to Decision Letter 0]

8 Apr 2024

Responses to the Academic Editor from the authors 

Thank you very much for taking the time to review our manuscript “Comprehensive transcriptomic analysis of prostate cancer lung metastases”, (Manuscript ID: PONE-D-23-41725). We appreciate the academic editor and reviewer's warm thoughts and comments and we already provided and uploaded rebuttal letters for each reviewer separately. Please find the detailed point by point responses to the academic editor below. The corresponding corrections highlighted with the tracked changes in the revised manuscript and re-submitted files accordingly.

Your sincerely,

The Authors

Point by Point Response to the Academic Editor:

Response 1. We appreciate this precious this comment. In order to follow the instruction, we have already re-checked our manuscript according to your journal style requirement, we have now made few changes inside the manuscript with tracked changes in the “revised manuscript” version accordingly. Furthermore, all figures have been already qualified and enhanced using “PACE” software accordingly.

2. Thank you for stating the following financial disclosure: "This work was funded by Deutsche Forschungsgemeinschaft (DFG) Schwerpunktprogramm µBone SPP2084 (to VS and SP). Duan Kang was supported by China Scholarship Council at the University of Luebeck (No. 202008440263)."

Please state what role the funders took in the study. If the funders had no role, please state: ""The funders had no role in study design, data collection and analysis, decision to publish, or preparation of the manuscript."" If this statement is not correct you must amend it as needed. Please include this amended Role of Funder statement in your cover letter; we will change the online submission form on your behalf. 

Response 2. We again appreciate this comment. In order to follow the instruction, we have already inserted the amended statement in our revised manuscript with tracked changes according to your instruction as following: 

3. Please include captions for your Supporting Information files at the end of your manuscript, and update any in-text citations to match accordingly. Please see our Supporting Information guidelines for more information: http://journals.plos.org/plosone/s/supporting-information..

Response 3. We again thank for this comment. In order to follow the instruction according the guidelines, we have already inserted captions for the supporting information (Supplementary file 1 and 2) in our revised manuscript highlighted with tracked changes.

Your sincerely,

The Authors

---

## [Decision Letter · Decision Letter 1]

7 May 2024

PONE-D-23-41725R1Comprehensive transcriptomic analysis of prostate cancer lung metastasesPLOS ONE

Dear Dr. Saraji,

Thank you for submitting your manuscript to PLOS ONE. After careful consideration, we feel that it has merit but does not fully meet PLOS ONE’s publication criteria as it currently stands. Therefore, we invite you to submit a revised version of the manuscript that addresses the points raised during the review process.

We look forward to receiving your revised manuscript.

Kind regards,

Sumit Kumar Hira, Ph.D.

Academic Editor

PLOS ONE

**Additional Editor Comments:**

Despite its potential significance, the manuscript fails to meet the standards required for publication in its present form. Unfortunately, the manuscript lacks the appropriate statistical methodologies and analyses required to support its claims effectively.

Reviewers' comments:

Reviewer's Responses to Questions

**Comments to the Author**

1. If the authors have adequately addressed your comments raised in a previous round of review and you feel that this manuscript is now acceptable for publication, you may indicate that here to bypass the “Comments to the Author” section, enter your conflict of interest statement in the “Confidential to Editor” section, and submit your "Accept" recommendation.

Reviewer #1: (No Response)

Reviewer #2: All comments have been addressed

2. Is the manuscript technically sound, and do the data support the conclusions?

Reviewer #1: No

Reviewer #2: Yes

3. Has the statistical analysis been performed appropriately and rigorously? 

Reviewer #1: No

Reviewer #2: Yes

4. Have the authors made all data underlying the findings in their manuscript fully available?

Reviewer #1: No

Reviewer #2: Yes

5. Is the manuscript presented in an intelligible fashion and written in standard English?

Reviewer #1: Yes

Reviewer #2: Yes

6. Review Comments to the Author

Reviewer #1: The authors had not addressed the points regarding statistical analysis which is the most important criteria for such transcriptome analysis.

Reviewer #2: (No Response)

7. PLOS authors have the option to publish the peer review history of their article (what does this mean?). If published, this will include your full peer review and any attached files.

Reviewer #1: No

Reviewer #2: No

---

## [Author Response · Author response to Decision Letter 1]

5 Jun 2024

Responses to the respected reviewer One

(Manuscript ID: PONE-D-23-41725, “Comprehensive transcriptomic analysis of prostate cancer lung metastases”

Dear Reviewer One,

Thank you very much for taking the time to review this manuscript. We sincerely appreciate your warm thoughts and comments. Here are the comprehensive point-by-point answers to your comments, accompanied with the highlighted tracked changes in the revised manuscript and re-submitted files where appropriate.

Your sincerely,

The Authors

Point by Point Response to the Reviewer #1: 

The study by Saraji et al. demonstrated the comprehensive transcriptomic analysis of prostate cancer lung metastases. The authors should modify the manuscript by considering following points.

Comment 1. The authors have used FFPE samples of prostate metastases samples and compared transcriptome profile of bone metastases and lung metastases samples. But authors have not used any control/adjacent control specimens to derive the specific differentially expressed genes between the subsets. Justify.

Response 1. We appreciate this precious reviewer's comment. We sincerely regret the unwelcome disparity! As a reference and control, primary PCa was accidentally reported as "PCa bone metastasis" in the result section, despite the fact that we had already noted this in the abstract and conclusion sections! Actually, for this study we have taken primary PCa as reference “setting primary prostate cancer” as a reference versus PCa lung metastases samples but meanwhile we also compared our results from PCa lung metastases to PCa bone metastasis and there we found that 209 genes were significantly upregulated, and 100 genes were significantly downregulated in prostate cancer lung metastases as we described in the result part. 

We apologize for the mistake in mentioning mentioned bone metastases instead of primary PCa! In order to correct the above-mentioned discrepancy and mistakes, we have now corrected and inserted the number of our primary PCa FFPE as control in the part “Method-material/Cohort” line 89.90 and also correction in the part “Result-1” in the lines 152 accordingly in the revised manuscript.

Comment 2. Regarding the statistical analysis of gene expression data, adjusted p value should be calculated to identify the differentially expressed genes in each category of samples.

Response 2. We thank the reviewer's comments. Given that the FDR (False discovery rate) technique improved and adjusted the features of the p-value distribution. Furthermore, FDR helped us filter out a subset of DEGs while preserving the genes of interest (supplementary table 1). However, one point we want to emphasise specifically is that if the FDR value > 0.05 and P value < 0.05, we will still use the p-value. Additionally, according to the Nanostring TM facilities while using nsolver to calculate all samples one can chose the FDR option output as the adjusted p-value. 

In order to answer this comment in another way, we also used the Benjamini-Hochberg’s method. Furthermore, we then used following formula to calculate the initial FDR, which is expressed as FDR and also correlated FDR in the supporting information table 2 (supplementary table 2).

FDR(x)=P(x)*m/x 

According to the value of X (rank value of p-value) from large to small, 

FDR(x)=min {FDR(x), FDR(x+1)}

We now created and uploaded a supplementary excel file named 2 represented our data based on the adjusted p value or FDR. Please refer to the supplementary 2 for the details.

Comment 3. mRNA count should be expressed as RPKM/FPKM which is the standard protocol for RNA sequencing data.

Response 3. We again appreciate this reviewer's comment. Nanostring TM is an alternative to traditional microarray technology and RNA-seq technology. The principle of Nanostring TM technology is based on the direct measurement of the barcode of the fluorescent molecule on the probe after the nucleic acid molecule hybridizes to the probe. 

According to the official manual of nCounter; nCounter Pro Analysis System User Manual (nanostring.com), we can use the companion software in order to obtain differentially expressed genes without converting the raw count expression matrix to RPKM/FPKM (e.g., E.Shenderov et al., 2023, PMID: 37012549). In addition, statistically speaking, regardless of whether FPKM is a necessary presentation method, we only used gene expression data in our experiments for differential analysis, and we did not use these data for survival analysis. However according to Nanostring again, variance analysis does not require FPKM normalization of the original data. In this study we only use gene expression matrix to analyse the expression differences of each gene (PMID: 37012549).

Comment 4. Here, RNA integrity with at least 90% of the fragments longer than 100 nucleotides were considered as suitable for gene expression analysis. Instead of the parameter, RIN should be calculated for the samples which is the standard protocol for gene expression analysis.

Response 4. We again appreciate this reviewer's comment. As we have also mentioned in the Response 4, according to the standard protocol provided by the manual nCounter Pro Analysis System User Manual (nanostring.com), we only need to measure the purity of mRNA and do not need to test the RNA integrity number to obtain samples that meet the standard criteria for the quality requirements. 

However, in our previous publication (Saraji et al., 202, PMID: 34529723) we have already reported a comparative and comprehensive analyses for RNA purity and integrity between classical bioanalyzers (Qubit TM, nanodrop®) and Nanostring TM and there we showed that in Nanostring TM the expressed genes which is produced by nCounter output already passed the RNA quality criteria including RIN, RNA binding density and RNA fragmentation threshold (Saraji et al., 202, PMID: 34529723).

Comment 5. The resolution of the figures are very poor. That should be modified. Response 5. We thank the reviewer for this attentive comment. In order to ensure the quality and clarity of the figures, we have now enhanced the resolution of our figures in a maximum possible quality (600dpi) and re-corrected the figures quality with using “PACE” software and re-uploaded them again on the online system accordingly.

Your sincerely,

The Authors

---

## [Decision Letter · Decision Letter 2]

20 Jun 2024

Comprehensive transcriptomic analysis of prostate cancer lung metastases

PONE-D-23-41725R2

Dear Dr. Saraji,

We’re pleased to inform you that your manuscript has been judged scientifically suitable for publication and will be formally accepted for publication once it meets all outstanding technical requirements.

Kind regards,

Sumit Kumar Hira, Ph.D.

Academic Editor

PLOS ONE

Additional Editor Comments (optional):

Reviewers' comments:

Reviewer's Responses to Questions

**Comments to the Author**

1. If the authors have adequately addressed your comments raised in a previous round of review and you feel that this manuscript is now acceptable for publication, you may indicate that here to bypass the “Comments to the Author” section, enter your conflict of interest statement in the “Confidential to Editor” section, and submit your "Accept" recommendation.

Reviewer #1: All comments have been addressed

2. Is the manuscript technically sound, and do the data support the conclusions?

Reviewer #1: Yes

3. Has the statistical analysis been performed appropriately and rigorously? 

Reviewer #1: Yes

4. Have the authors made all data underlying the findings in their manuscript fully available?

Reviewer #1: Yes

5. Is the manuscript presented in an intelligible fashion and written in standard English?

Reviewer #1: Yes

6. Review Comments to the Author

Reviewer #1: The authors have addressed the specific comment raised by the reviewers and the manuscript may be accepted in the revised version.

7. PLOS authors have the option to publish the peer review history of their article (what does this mean?). If published, this will include your full peer review and any attached files.

Reviewer #1: No

---

## [Editor Report · Acceptance letter]

27 Jun 2024

PONE-D-23-41725R2 

PLOS ONE

Dear Dr. Saraji, 

I'm pleased to inform you that your manuscript has been deemed suitable for publication in PLOS ONE. Congratulations! Your manuscript is now being handed over to our production team.

Kind regards, 

on behalf of

Dr. Sumit Kumar Hira 

Academic Editor

PLOS ONE